# Ham Tourism in Andalusia: An Untapped Opportunity in the Rural Environment

**DOI:** 10.3390/foods11152277

**Published:** 2022-07-30

**Authors:** Mª Genoveva Millán Vázquez de la Torre, José Luis Sánchez-Ollero, Mª Genoveva Dancausa Millán

**Affiliations:** 1Department of Quantitative Methods, Universidad Loyola Andalucía, 14004 Cordoba, Spain; gmillan@uloyola.es; 2Department of Applied Economics, University of Malaga, 29071 Malaga, Spain; 3Department of Statistics, University of Cordoba, Puerta Nueva s/n, 14071 Cordoba, Spain; z62damim@uco.es

**Keywords:** Iberian ham, Andalusia, factor analysis, gastronomic tourism

## Abstract

Iberian ham is a food product of great quality endorsed by Protected Denominations of Origin, which is beginning to be marketed as a tourist product, and visits to pastures, ham dryers, etc., are becoming tourist attractions in the rural areas of Andalusia. In this research, a factor analysis with VARIMAX rotation is carried out to determine the factors that influence the development of ham tourism. Five components are determined, of which the supply factor is one of the most important. Pull factors are highlighted as the local gastronomy and heritage, among others, and push factors include visiting dryers, pastures, and ham museums. Based on these results and the descriptive analysis of the profile of the ham tourist, it is found that the ham tourist is very satisfied with the visit made, and that the tourist would repeat the experience. However, it is observed that it is necessary to create routes combined with other better known gastronomic products, such as wine, and carry out marketing campaigns to publicize this tourist product in the national and international market, because at present it is local tourists who perform this type of tourism.

## 1. Introduction

After the COVID-19 pandemic and according to the Food Barometer [1], there is an increase in the awareness of Spaniards of eating a healthy diet, and three out of four Spaniards are more aware of the importance of following a healthy diet such as the Mediterranean diet, either by taking care of their health (88%) or by issues related to animal care and the environment (23%). In fact, 37% of respondents say they began to eat better after the most critical period of the pandemic. 

The Mediterranean diet, one of the fundamental pillars of Spanish food and recognized by UNESCO in 2013 as Intangible Cultural Heritage of Humanity [2], is a healthy diet [3,4,5] typical of the countries of the Mediterranean Basin (Cyprus, Croatia, Spain, Greece, Italy, Morocco, and Portugal). The Mediterranean diet has beneficial properties for health, especially due to the type of fats used (olive oil, fish, and nuts), the proportions of the main nutrients of the recipes (cereals and vegetables as the basis of dishes, and meats or similar as a “garnish”), the richness in micronutrients that the diet contains, and the use of fruit, seasonal vegetables, aromatic herbs, and condiments. The Mediterranean diet is balanced, complete [6], and consists of various nutrients. It is mainly based on consuming food obtained through traditional crops (such as wheat, olive, or vine).

The Mediterranean diet had its first antecedents in ancient Rome, with the expansion of the Roman Empire, and extended to the entire Mediterranean area. Since then, flour, bread, and olive oil have been used, in addition to many other products collected from nature, which are still used today in the preparation of culinary dishes.

Some of the most used nutrients in the Mediterranean diet are:-Olive oil: it can be said that it is the main element of Mediterranean cuisine, provides vitamins and antioxidants, and also serves as a source of fat. The main type of fat is oleic acid, which is very beneficial for health [7].-Meats: provide considerable amounts of protein, iron, vitamins, and minerals. Among the meats, cured meats stand out, such as Serrano/Iberian ham, and in Roman times the slaughter of the pig was already carried out. There were special cooks called “vicarius supra cenas” who prepared the meats. The most precious part was the ham (back leg of the pig), and it could be consumed only by the wealthiest inhabitants. This food is a good source of protein, and it is rich in minerals (iron and zinc), phosphorus, potassium, sodium, calcium, protein, and vitamins. It has unsaturated fats, of which oleic acid can be underlined; therefore, it has properties similar to those of olive oil, an essential element in the Mediterranean diet.-Fruits: they provide a good supply of vitamins, fibre, water, and minerals. It should also be noted that these foods have few calories. Fruits must be included in any diet for it to be considered stable or balanced.-Vegetables: this family of nutrients has a good amount of fibre, vitamins, minerals, and carbohydrates, although it almost completely lacks protein and fat.

Of all the nutrients mentioned above, in this research work we analysed Iberian ham as an element of Andalusian gastronomy, especially from a tourist attraction perspective, with the aim of determining the profile of the demand for Iberian ham tourism and the factors that characterize this segment of tourism. For this, univariate descriptive statistics were used, which allowed us to determine the most significant percentages of each variable, as was factor analysis, which allowed us to obtain the five most relevant factors that influence ham tourism. Likewise, we carried out a correlation analysis to discover the most important relationships between the variables.

The structure of the work is based on the analysis of the denominations of origin of Andalusian ham, carrying out fieldwork of 409 surveys, to which the aforementioned techniques were applied, obtaining as a result a high degree of satisfaction on the part of the tourist. The desire to repeat the experience was also analysed, with the contribution of this work to the scientific literature being the determination of the factors that influence Iberian ham tourism.

## 2. The PDOs of Iberian Ham

The gastronomy of a region plays a fundamental role when it comes to attracting people from other places with the aim of knowing it and being able to taste it. The Andalusia region, located in the south of Spain, has a vast gastronomy formed by recipes inherited from previous centuries, being sometimes a fusion of peoples that cohabited the region, such as the Jewish, Arab, and Christian cultures, and based on top quality foods such as oil, wine, or ham that are endorsed by denominations of origin that certify the quality of these products.

Spain is part of the European Union and therefore cannot dictate rules of denomination without the consent of the European Union; thus, in 2006, the European Commission developed a European policy that recognizes and protects the denominations of certain specific products that are related to a territory or a production method. This recognition translates into quality logos that make it possible to identify products of differentiated quality in the EU and that, through specific controls, also guarantee their authenticity [8].

Two of them have geographical connotations, namely, the Protected Designation of Origin (PDO) and the Protected Geographical Indication (PGI), and the third relates to traditional production methods (Traditional Specialties Guaranteed (TSG)).

The difference between PDO and PGI is that the first is more demanding, as the production, processing, and elaboration are carried out in the same geographical area. Examples in this category include the olive oil of Priego de Córdoba, the wine of Jerez, and the Honey of Alcarria. However, with respect to products in the PGI category, it is not mandatory that all the phases are carried out in the same geographical area (Cordero Manchego, Cea bread, etc.). 

Spain has 199 appellations of origin and 152 protected geographical indications [9] (Table 1), of which 28 PDOs and 31 IGPs are in Andalusia, representing 14.07% and 20.39%, respectively (Table 1). Analysing by products, those that correspond to the pig are included in the meat products section (cooked, salted, smoked, etc.).

According to data from the MAPA [9], in the months of August 2020 to July 2021, sales of agri-food products abroad reached 57,000 million euros, among which the Iberian sector played a very important role.

The Iberian products (hams, loin, sausage sausages) exceeded 2000 million euros, accounting for 8% of the total of the Spanish meat industry, which, in turn, is the fourth industrial sector in the country, with a business value of 27,950 million euros (22.2% of the entire Spanish agri-food sector).

Within the Iberian sector, Iberian ham stands out par excellence and has a quality increasingly demanded by consumers, while it is becoming known in international markets due to its beneficial properties for health. The knowledge and experience associated with the cultural heritage of specific territories have led to the incorporation of new technologies, resulting in production of the highest quality, responding to increasingly requested sustainability standards.

Iberian hams are foods that are oriented towards excellence and are highly valued in Spain, but also by consumers from other EU countries, such as Germany, which imports more than 11,803 tons (more than 100 million euros), France (10,406 tons, 99.67 million euros), Italy, and Portugal. Also noteworthy are exports to other countries, such as the United Kingdom, the United States, Mexico, and Japan, in addition to China (Figure 1 and Figure 2).

In 2021, the Interprofessional Association of the Iberian Pig (ASICI) and the European Union launched the campaign, “Hams of Spain, ambassadors of Europe in the world”. The objective was to consolidate internationally the culinary emblem that is the Iberian ham as a unique gourmet product in the world, reaching more than 455 million potential consumers, managing to appear in 917 articles of various media, and creating a virtual community of 270,000 followers. This campaign was part of a training, information, and promotion plan around Iberian Ham intended to advance the consolidation of this gourmet product in strategic markets such as France, Germany, Mexico, China, and Spain itself. For the development of this plan, dissemination activities were carried out, such as educational workshops, training events in hospitality schools, experiential workshops, international cutting competitions, and trips to production areas. In this sense, it was a campaign that added to the efforts also made by the Ministry of Agriculture, Fisheries, and Food in its Food Strategy of Spain and its specific campaign, "The richest country in the world", and its Food Award of Spain for the best Ham.

Through these information campaigns, Iberian ham was made known as an essential element of Spanish gastronomy; the different products from the Iberian pig and how to recognise them were also emphasised.

However, a non-expert consumer may include all varieties of ham (Serrano ham and Iberian ham) within Iberian ham. The difference between Serrano ham and Iberian ham is that Serrano ham comes from a white pig that can be found in other countries, while Iberian ham comes from Iberian pigs, which are native to the Iberian Peninsula and that have unique characteristics that give the products a higher quality (Figure 3).

The Iberian pig develops a fat that infiltrates subcutaneous (bacon), intermuscular (lard and others), and intramuscular (veining) deposits. The composition of these fats is fundamentally determined by the pig’s diet, resulting in pigs fed in the *montanera* (season of maturation of the acorns and fattening phase of the pig by means of acorns [10]) having a high content of monounsaturated and polyunsaturated fats [11], while others have a very low melting point (between 32 °C and 36 °C). This is a very important detail to take into account, since it is responsible for the good diffusion of aromas that are fat-soluble, which when combined with high external temperatures, leads to increases in aromas and flavours. It is a highly recommended food for any type of balanced diet (Figure 4), being considered one of the healthiest foods in the Mediterranean diet. One hundred grams of Iberian ham provide 24% of daily consumption of vitamins of type B and 30% of protein, which is essential for growth. In addition, it carries all the essential amino acids that the body cannot synthesize, so it can be said that Iberian ham collaborates in the correct functioning of human metabolism. Its fat and salt contents are not high, since 65% of the ham is water; it is a low-calorie food, so it can be very convenient for hypocaloric diets.
-Fruits: they provide a good amount of vitamins, fibre, water, and minerals; it should also be noted that these foods have few calories. In any diet, fruits must be included for it to be stable or balanced.-Vegetables: this family of nutrients has a good amount of fibre, vitamins, minerals, and carbohydrates, although they almost completely lack protein and fat.

Of all the nutrients mentioned above, in this research work we analyse Iberian ham as an element of Andalusian gastronomy, including from a tourist attraction perspective, to know the factors that characterize the ham tourist, performing a factor analysis to determine them.

The sales denomination of the products produced from the Iberian pig is obligatorily composed of three designations: type of product, breed, and type of feed.
Product type designation: ham, shoulder, chorizo, sausage, loin cane or stuffed loin or loin.Racial designation: Iberian or pure Iberian (when the product is obtained from pigs whose parents are pure Iberian breeders).Feed type designation:
Acorn or finished in *montanera*: for those products obtained from pigs that are destined for slaughter immediately after the exclusive use of acorns, grass, and other natural resources of the pasture without the possibility of feed administration.Of recebo or finished in *recebo*: for those products obtained from pigs that after replenishing a minimum of weight in *montanera,* their diet until the moment of their slaughter is completed by the contribution of feed.Field bait: for those products obtained from pigs whose feed is based on feed and which complete their feeding by a minimum stay of 60 days in the field prior to slaughter, during which they also receive a feed-based diet.Bait: obtained from animals whose feeding until the moment of slaughter is based on feed.

In Spain there are only four PDOs of Iberian ham (product type designation and racial designation), of which two are in Andalusia, namely, the Iberian ham of “Pedroches” located in the province of Córdoba, and the Jabugo located in the province of Huelva. The main objective of the Protected Designations of Origin of Ham is to protect production linked to a delimited and specific geographical area, becoming an important instrument of economic, social, and environmental development of the area [10]. It is also a tool to help and defend the consumer against imitations and abuses in order to guarantee authenticity, avoid fraud, and offer the consumer correct information backed by years of tradition and the local and differentiated characteristics due to the geographical environment of the *dehesa* (meadow), the racial purity of the Iberian pig, and a characteristic climate that differentiates the elaboration process. This protection has an impact on the greater added value of the products covered by a PDO and, therefore, greater added value of the final products, which affects the entire production chain, reaching the farmer and promoting, in addition, the name of a territory that joins a gastronomic product of the highest quality [13].

The differences between the different Denominations of Origin of Iberian ham are based mainly on the place where the pigs are raised and fattened, where their products are made, and the scope of their certification in terms of the typologies they certify [9].
*PDO Guijuelo:* It is the first Protected Designation of Origin of Iberian ham (year 1986). It is characterized because the origin of the pigs can be from anywhere in Spain, and as for the drying processes, these take place in the province of Salamanca (seventy-eight municipalities located in the southeast of Salamanca in the middle of the Salamanca pasture) at high altitude (more than 1000 m), so the process requires less salt, and thus softer hams are obtained. This denomination certifies the categories of acorn and field bait, and the breed must be 100% Iberian or 75% Iberian.*PDO Dehesa de Extremadura:* Recognized by the European Union as a PDO in 1996. In this Denomination of Origin, all phases are carried out in the Autonomous Community of Extremadura, from the breeding of piglets (young pigs) in the meadows of holm oaks and cork oaks present in all the municipalities of the Autonomous Community of Extremadura, to the curing of hams and shoulders. The categories it certifies are acorn and field bait, and the breed must be 100% Iberian or 75% Iberian.*PDO Jabugo* (formerly Huelva ham): Recognized as a PDO in 1995. Sus hams are known for the curing process in natural wineries that meet very specific microclimate conditions; although the pigs can come from any area of Spain, the curing must be done in the province of Huelva, in any of the 37 municipalities located in the region of La Sierra de Huelva. This designation of origin only certifies 100% Iberian acorn pigs.*PDO Los Pedroches:* The Denomination of Origin Los Pedroches is the youngest of the four that covers Iberian pork hams and shoulders (registered in the year 2010). It integrates a total of 32 municipalities of the Valley of the Pedroches, the Guadiato Valley, and others of the Sierra de Córdoba with an elevation of more than 300 m, all of them located north of the province of Córdoba. Therefore, each and every one of the pieces, both the hams and the shoulders (front leg of the pig) protected by the Protected Designation of Origin Los Pedroches, go on the market identified by the Regulatory Council through an inviolable seal and a label, both numbered individually, in which the category of the piece is specified. The seal is placed at the time of slaughter of the pigs and the label at the time of dispatch of the product, once all quality controls have been passed. Each piece is identified one by one because it is unique and is easily recognizable by the colour of the vitola: black for 100% Iberian Acorn; red for Iberian Acorn (75% Iberian Breed), and green for 100% Iberian Field Bait or Iberian Field Bait (75% Iberian Breed). All phases of the process must be carried out in the Region of Los Pedroches, from the birth of the piglets to the curing of the ham. This curing differs from that of the other areas by the hot summer that accentuates the flavours and aromas of the hams and shoulders to reach unparalleled sweetness. This Denomination of Origin certifies 100% Iberian animals exclusively, of both the categories of acorn and field bait, although more than 95% of the pigs protected each year are acorns.

Figure 5 shows the geographical distribution of the denominations of origin and PGI of the ham of Spain. In Andalusia there are the PODs of Iberian ham of Los Pedroches and Jabugo and the PGI of White ham of Trévelez and ham of Serón; therefore, this region has a quality product, endorsed by the PDO and PGI, which link the raw material of a food such as Iberian ham with its territory, making it a potential tourist resource, which can be exploited by creating gastronomic routes around that key product that characterizes them. They usually offer related activities such as visits to pastures and ham museums, tastings, workshops on the cutting of ham and the preparation of dishes related to ham, etc., which allow the autochthonous productive culture to be consolidated and the regional products re-evaluated [14]. This boosts the regional economy, with gastronomic itineraries being an instrument to position the ham and associate it with a geographical quality appellation [15,16]. However, to configure a gastronomic route of Iberian ham, different actors must participate, such as producers, merchants, restaurateurs, and hoteliers, as well as public bodies such as the Provincial Councils or the City Councils, the union of synergies of all of them being fundamental for the good development of the route [17].

At present, the routes of Iberian ham are little commercialized but have great potential, since various types of tourism can be developed in them, such as gastronomic tourism, rural tourism, ecotourism, etc. Not only can one visit ham dryers, which are the places where Iberian hams are manufactured, but also the pastures that are unique ecosystems in the world, being a sustainable model of exploitation and conservation, formed by two million hectares of natural ecosystems distributed throughout the provinces of Salamanca, Caceres, Badajoz, Huelva, Seville, Córdoba, and Ciudad Real. In this Mediterranean forest, with acidic soils and scattered trees, especially holm oaks and cork oaks, with thickets such as rockroses and brooms, we find wild fauna (deer, wild boar, or dove) and native domesticated animals, such as the Iberian pig, the *retinta* cow, and the *merino* sheep, which can be attractions for the city tourist. In addition, the PDO of the Pedroches, which also has nearby natural parks such as Cardeña-Montoro, would attract rural tourists who could take the ham route.

## 3. Literature Review

Tasting the gastronomy of a place can be considered as a socio-psychological experience [18,19,20,21,22]. Today tourists travel in search of sensations and new experiences, not only the visual but also those of the senses such as taste or smell [23]. In tourism, especially cultural tourism, traditionally one of the senses, the visual, has been prioritized over the rest of them. In fact, sometimes the tourist becomes a simple observer of reality who contemplates a painting, an architectural work, or a landscape, not using enough of the other senses during the trip. However, more and more tourists are beginning to demand trips where they can also involve other senses more deeply, and thus they seek the need to appreciate the atmosphere of the place, to enjoy the local food, to know the customs of the place, and to participate in a certain event such as a ham festival. At present there are tourist activities where one can enjoy other senses, such as oleotourism, wine tourism, and ham tourism, where one can gain complete sensory experiences [24] since the visitor can experience the pleasure of taste, smell, touch, sight, and sound. 

However, not all tourists experience the same sensations, since these depend on factors such as socio-demographic characteristics and subjective experience. In this context, motivation and satisfaction are two essential elements that determine individual behaviour in the field of tourism [25,26,27,28], explaining why a person travels and seeks to meet their needs through different experiences. Therefore, motivations are directly related to satisfaction [29].

The scientific literature on the motivation of tourism tries to explain the reason why tourists decide to visit that destination instead of another place, the type of experience they want to receive, and the types of activities they want to do [30]. This is because they are “pushed” to travel for internal reasons or factors, or because they are “attracted” by the attributes of a destination. Push factors are more related to internal or emotional aspects [31], such as the desire to escape [32,33], rest and relaxation [34], adventure [35], cultural experience [36], or social interaction [37]. Pull factors are linked to external, situational, or cognitive aspects, such as the attributes of the chosen destination, the leisure infrastructure, or cultural or natural characteristics [38]. However, these attributes of a destination can reinforce pushing motivations [39]. 

Therefore, motivation is an essential element that affects travel behaviour and determines different aspects of the tourist activity with respect to (1) the reasons for traveling or why, (2) the specific destination or where, and (3) the results obtained or the overall satisfaction with the trip [40].

There are multiple studies that analyse the relationship between motivation and satisfaction in the tourism sector from different perspectives and work methodologies [41,42,43,44] and in different market sectors [45,46,47,48], as well as studies on gastronomic routes [49,50].

In research on gastronomic tourism, the most expert tourists in the field (they choose their gastronomy as the main motivation of the destination) obtain a higher level of satisfaction than tourists who chose gastronomy as a secondary reason for making the trip. They start from the hypothesis that organized gastronomy tours work like a well-oiled machine. Food tourists have specific needs, and tour organizers know from experience what tourists expect and need. On the other hand, gastronomic tourists know what to expect, because their main source of information is personal and not commercial. Thus, the gap between the expectation and the experience of tourists is reduced to a minimum. However, this is only speculation rather than research, and new and better data are needed to fully understand this difference in satisfaction between the gastronomic tourist and the general tourist.

It can be deduced that the satisfaction of the tourist in general as his intention to repeat the trip in the future are partially determined by the tourist’s assessment of the different attributes of the destination [51]. In this sense, many studies explore the performance of a destination by analysing tourism satisfaction in different aspects of the place [52,53,54]. In addition, research on loyalty to a destination shows that one of the most decisive factors in a new visit of tourists to the area is their satisfaction with previous stays [55,56,57].

Other research analyses food according to the territory in countries such as Portugal [58], Japan [59,60], China [61], India [62,63], Croatia [64], Greece [65,66], France [67,68,69], Hungary [70], and Poland [71]; or depending on a specific food such as oil [72], wine [73], cheese [74], tuna fish [75], cod [76], and cider [77]; or depending on the gastronomic routes [78,79]. Other studies analyse gastronomic destinations based on the dishes prepared according to traditional recipes, such as pizza [80,81], salmorejo and rabo de toro (oxtail) [82], plato minero (miner’s dish) [83], and noodles [84], or depending on the life cycle of the gastronomic product. The basic concept of the life cycle in tourism studies, attributed to Butler [85], refers to the fact that tourist destinations usually show an evolutionary path made up of different stages: exploration, participation, development, consolidation, and stagnation and decline or rejuvenation. The life cycle model in a tourism destination (LCDT) is not a predictive or explanatory theory but rather a generalization based on observations that have high value for the strategic planning of tourism. When applying the LCDT model to gastronomic tourism, it can be said that the destination depends on the attraction capacity of the food/raw materials with the place in terms of wineries, meadows, ham dryers, restaurants, gastronomic festivals, etc., in such a way that if these elements evolve, so will the tourist destination. It can be thought, however, that other types of attractions not related to gastronomy will also affect the evolution of the destination.

As Butler [86] argues, there are a number of reasons why the LCDT model remains popular and useful, both theoretically and practically. First of all, it is clear that tourist areas are dynamic. Change is usually brought about by a combination of triggering factors, either external factors (especially competition or chaos) or deliberate political, business initiatives. Some gastronomic tourism regions are strategically marketed and promoted by the destination and the tourist organizations of the country, although in general it is usually the action of the entrepreneur/manager of gastronomic routes/PDOs that initiates and sustains the development process. In the process of changing a destination there are usually identifiable stages. For example, of the gastronomic tourism destinations, there are some that are in their infancy, such as oil tourism [87], or highly developed, such as the wine routes in the Napa Valley [88]. The spatial components of the CVDT model have not been studied in depth, but Butler argued that tourism spreads geographically over time. This, within a gastronomic tourism region, could be directly related to the expansion of Michelin star restaurants, PDOs, the influence of gastronomic routes on tourist traffic, and the location of visitor services. Likewise, the location and accessibility of the destination with respect to the origin of gastronomic tourists is of vital importance.

In any application of a life cycle model, temporal indicators must be considered (that is, what changes over time: the volume of tourism, the number of restaurants or wine cellars, dryers, oil mills, the number of activities related to gastronomy, etc.?), as well as the spatial dimension. The life cycle concept has been frequently used and criticized in the tourism literature. In fact, two volumes of articles referring to the life cycle of tourist destinations have been published in book form [86,89].

However, this model is useful to conceptualize development as a growth process, especially if the development is driven by demand, and in the alert of potential decline [90], and in specific cases such as the wine routes in Portugal [91], or different case studies comparing the phase of the life cycle where gastronomy tourism is found, such as in the Lion and the Rhône–Alpes region [92].

Therefore it can be concluded that in the analysis of the literature, it can be seen that there are multiple investigations of gastronomic destinations depending on the characteristic that is to be analysed; from different points of view, in this review of the literature, some of them have been taken into account that have served as the basis for the elaboration of this research, especially those based on specific gastronomic products, such as wine or oil [93,94].

## 4. Materials and Methods

A survey was carried out with the population composed of tourist consumers who visited any of the denominations of origin of the ham or conducted an Iberian ham route of Andalusia from September 2021 to January 2022. To select the sample, the simple random sampling technique was used, in which all tourists had the same probability of being selected to be part of the sample, and a sample size of 409 individuals was chosen based on the formula for calculating sample size for unknown or infinite populations, with the aim of knowing what factors influence the ham tourist. In order for the questionnaire to be valid as a measurement instrument and according to [95,96], a pre-test was carried out on 35 gastronomic tourists to verify that it met the following characteristics:Be simple, viable, and accepted by tourists and researchers (feasibility).Be reliable and precise, that is, with error-free measurements (reliability–consistency). To verify this, Cronbach’s alpha was used.Be suitable for the problem to be measured (content validity).Reflect the underlying theory in the phenomenon or concept to be measured (construct validity). Questionnaires like those used by gastronomic tourism researchers [7,93,94,97] were used.Be able to measure changes, both in different individuals and in the responses of the same individual over time (sensitivity to change).

For this, a questionnaire was carried out consisting of 38 questions divided into 4 blocks (Table 2). The first block was used to collect personal information from tourists (age, sex, level of education, marital status, etc.), while the second block contained information about the route taken (how the ham route was known, if the route covered their expectations, what would improve, if they came expressly to make the gastronomic route, etc.). The third block was used to record the motivation to perform ham tourism (the reason for the gastronomic route and use and consumption of ham, as well as assessments regarding ham, texture, flavour, etc.). The fourth block of assessment was used for the services received during the route, the price of the trip, the hospitality and treatment received, etc. Information was collected using a questionnaire directed at the population of tourist consumers visiting a ham route/POD Iberian ham in Andalusia. Regarding the type of questions used, some of them were of a nominal qualitative type, such as gender, marital status, etc.; others were of an ordinal qualitative type, such as level of education and other quantitative questions such as income level; finally, others were tabulated on a Likert scale from 1 to 10 as the assessment of the restoration, accommodation, security of the destination, heritage, etc.

The access by the surveyors to the ham route (dryers, restaurants, ham museums, etc.) and the conduct of interviews with tourists were authorized by the managing body and owner of the POD.

Prior to the completion of the questionnaire, tourists were informed of the academic purposes and their anonymity in answering. Consent to take the questionnaire was verbal. At all times, the visitor’s anonymity with respect to the ham route/POD was guaranteed.

With the information obtained in the survey, the following was carried out:The model used was Baloglu and McCleary’s Path Model of the determinants of Tourism Destination Image before actual visitation [98], which refers to two forces that influence the image of a tourist destination: push–pull factors result from the breakdown of two motivational forces. The pushes explain the desire to travel, are sociopsychological factors, and work by impulse, while the pulls result from external forces and depend on the evaluations made to the attributes of the destination (tangible resources: pastures, ham dryers, ham museums, ham festivals, gastronomy; and/or intangibles: environment of the place, traditions), and they are the attractions that motivate the choice of destination and/or tourist product. In this model, two basic perspectives of motivation analysis are proposed: motivation as an impulse, of instinctive origin, based on internal needs such as the need for knowledge of oil manufacturing; and motivation as attraction, based on reason and emotion. In this research, we tried to analyse the factors that characterize ham tourism, and a questionnaire composed of four blocks was used.Block 1. First, sociodemographic data were collected:SexAgePlace of residenceEducationMarital statusWorking situationMonthly rentBlock 2. The second part of the questionnaire was used to collect data on tourism behaviour, including both qualitative and quantitative data:Mode of transportHow the destination was selectedTravel organizationContracted servicesDaily cost of hosting services (quantitative)Travel group compositionDurationType of accommodationWhy will not stay more days in AndalusiaVisit to other citiesInternational daily expenditure per person (quantitative)Previous visitsMain motivation to travel to AndalusiaBasic motivation to travel to AndalusiaBlock 3. The third part of the questionnaire adopted an importance-assessment approach. Specifically, visitors were asked to measure the levels of importance they attributed to certain items (including both push factors and pull factors) and then the degree to which they could express their satisfaction with them. Specifically, 19 items were included that could be divided into two blocks. The first block of external factors of attraction included the following:Historical and monumental heritageLocal gastronomyEvening entertainmentConservation of the city’s environmentCleanliness of the cityEase of access—communications, roads, etc.TelecommunicationsPublic transportTourist information and signageCitizen securityFriendliness of the peopleValue for money accommodationValue for money restaurantsSecond block of internal pull or push factorsVisit ham dryersVisit ham museumsVisit pasturesVisit cultural or historical places or eventsEnjoy natureGetting to know a different culture

The identification of the items was determined according to the literature review and by including and adapting some elements from the observation, and from the discussions held with different key informants, to make a total adaptation of the population and the attributes of the destination.

In order to measure the levels of importance and to measure the levels of satisfaction, a ten-point Likert scale was used, where ten represented the highest levels according to the items.

Since satisfaction with a particular destination is more than visitor satisfaction with the services used and the attributes of the destination [99,100], a measure containing the level of overall satisfaction obtained with the visit was also included. This measure reinforced the comprehensive (holistic) approach being carried out in this study.

Block 4. Finally, respondents were asked to indicate whether they would return to Andalusia, and whether they would recommend the trip to others.

The treatment of the questionnaires obtained was carried out by processing the coded answers.2.To examine the degree to which the defined indicators adequately measured the concept (construct) to be measured, an exploratory factor analysis was carried out, a technique that allows a reduction in the dimensionality of the data based on the analysis of the correlation between the variables. The KMO (Kaiser–Meyer–Olkin) coefficient was previously applied with the aim of determining that the factor analysis procedure that was carried out was pertinent. This statistic varied between 0 and 1. It is commonly accepted that:If KMO < 0.5, it would not be acceptable to do a factor analysis.If 0.5 < KMO < 0.6 is the degree of mean correlation, there would be mean acceptance in the results of the factor analysis.KMO > 0.7 indicates high correlation and, therefore, convenience of a factor analysis.

Bartlett’s sphericity test was also calculated to verify that the correlation matrix of the defined factors was not an identity matrix, which implies no correlation between the variables A level of significance greater than 0.05 does not ensure that the factorial model is suitable for explaining the data because the null hypothesis of sphericity cannot be rejected.

The Statistical Pearson Correlation Coefficient was applied to determine the correlation between factor loads and the overall satisfaction variable. The determination of the reliability of the instrument was made using Cronbach’s alpha coefficient, which is an index of internal consistency; values ranging between 0 and 1 serve to check if it is a reliable instrument for making measurements. Its interpretation as a statistic is that as it approaches 1, the better the reliability, and acceptable reliability is considered to begin from 0.70 [101,102,103].

## 5. Results

Table 3 shows the most significant results of the variables analysed. The general profile of the tourist who makes a visit to a ham dryer, pasture, ham museum etc., was determined to be a person aged between 50 and 59 years (31.1%), with secondary education (43.8%), married (47.7%), and a medium income level (€1500–2000), a profile very similar to the gastronomic tourist who visits the Jerte Valley or the Trujillo cheese festival [104], but which differs from the gastronomic tourist of other countries such as the Dominican Republic, where tourists are younger, under 30 years old (38.3%), and gastronomic motivations are not the main reason to travel [105].

Another characteristic of the ham tourist was that 58.9% of the ham tourism was local tourism, partly because the ham routes are not very commercialized in international markets and also because after the pandemic tourists make trips closer to their places of residence and in contact with nature; therefore, it is a tourist who does not spend the night in the place (53.5%) and is aware of the product (ham), since the tourist consumes it habitually (49.4%), because it is part of his diet; 66.7% considered themselves interested in ham, and their opinion regarding the texture and flavour considered it good, with their average daily expenditure being between €66 and 100 (42.1%); some tourists had higher expenses because in the ham dryers they bought some Iberian product, the visit to ham dryers being the preferred place for tourists as well as the visit to pastures (50.4%). The degree of satisfaction with the route was greater than 75% in 94.1% of cases, and they would repeat that same experience 98.1%, these same tourists being a potential future demand.

Table 4 presents the results of the KMO test and Bartlett’s sphericity test; the value of the latter was represented by the Chi-square statistic, a value that was high enough (4869.413) to recommend entering the factor analysis test, and it also presented a perfect significance of zero value, which allowed us to reject the null hypothesis. On the other hand, the value of the KMO, which measures the degree of adequacy of the sample, was KMO = 0.75385, which indicated that factor analysis was appropriate.

Therefore, the questionnaire was subjected to a factor analysis of main components with Varimax rotation, a method that minimizes the number of variables with high load in each component, thus improving the ability to interpret factors.

The matrix of communalities (Table 5) explained the percentage of the variance of the phenomenon that manifested each variable. In this case, all the variables had high contributions, greater than 0.5, which demonstrated the high capacity of the common factors to explain the variability of each variable. It could be verified that the attributes monthly income, average daily expenditure, valuation of accommodation, restoration valuation, valuation of heritage, and use of olive oil (frequency of consumption) were the highest values of the matrix, so their participation in the analysis of the resulting components would be greater.

Figure 6 presents the sedimentation results with the number of factors that provide the best explanation of the object of study. As can be seen, the number of factors or components appears on the X axis, which coincides with the number of items; on the Y axis are represented the eigenvalues, which are equivalent to the variance explained by each factor. The cut-off point to establish the number of factors that are chosen as sufficient is located at inflection point 4 of the descending line that joins the various eigenvalues.

When the previous results compared with the matrix of the total variance were explained (Table 6), it was verified that with the first five components, 72.781% of the total variability of the analysed phenomenon was explained.

After the rotation of the factors, a factorial matrix was obtained that showed the way in which the variables were grouped with respect to a certain factor, and which were the ones that saturated it the most. The weights or loads of the variables that made up a factor expressed the importance of each variable for each component in question, and they could even serve to identify the factor. Table 7 shows the factors, as well as the variables that carried the most load in each one. Variables with correlation values greater than 0.6 were preserved. The factors identified were the following: factor 1, which can be identified as “offer”, factor 2 as “personal”, factor 3 as “ham relationship”, factor 4 as “leisure and safety”, and factor 5 as “satisfaction”.

It is clear that the sets of variables that were formed for the first presentation of the questionnaire were modified to some extent. From the groups formed in the matrix, it is possible to identify each component according to the concept that these variables measure.

Table 7 shows the five components obtained and in different colours the variables that are part of each component.

On the other hand, in component 1 (offer), the variables that provided ratings on the offer were more important, so this was named “offer” (Table 7); this first factor explained 24.583% of the variance (Table 6) and was formed by seven items, which were accommodation valuation (0.889), quality assessment of ham tourism offer (0.889), restoration valuation (0.875), valuation of assets (0.835), valuation of route information (0.704), assessment of attention and treatment received (0.695), and valuation of synthetic index of perception (0.629), which were within the pull factors.

In the case of component 2 (personnel), as shown in the matrix (Table 7), it explained 16.235 of the variance (Table 6), and its most important variables were the related travel information. They obtained values greater than 0.9, namely, personal income (0.93) and average daily expenditure (0.91), and above 0.75 the level of education (0.795) and level of studies and duration of travel (0.755).

In this sense, this set could maintain the name of personnel, since the variables with the highest degree of contribution were those that provided information on the economic information of the ham tourist, some of them related to the trip, such as the average daily expenditure and the duration of the trip.

The results of the matrix show that in the third component, “relationship with the ham”, presenting a high load of the variables related to the route were valuations of the leisure offer, signalling, and safety of the route.

The third dimension, “ham”, which explained 12.53% of the variance, was formed by three items that relate the knowledge and consumption of ham of the ham tourist (Figure 7). Where a high relationship was observed between the classification of the ham tourist (ham lovers, connoisseurs in ham, interested in ham, and initiated in ham), the type of Iberian ham (acorn, field bait, bait, and recebo) and the number of times the tourist consumes ham, either as an aperitif or in the preparation of dishes such as *flamenquín*, thicker lines indicate more relationships, so the lovers and connoisseurs of the ham usually eat ham almost every day and eat ham of better Iberian quality; those interested in the ham, which were the majority of tourists to come from the autonomous community, knew the product. They usually consumed the ham received several times a week, while the tourist with less knowledge of the product consumed less ham and ham of inferior quality.

The fourth component (leisure and security) explained 12.09% of the variance and was made up of three variables: leisure/entertainment offer rating (0.819), assessment of tourist signage of the route/PDO (0.748), and citizen security assessment of the route (0.748).

The fifth component (satisfaction) explained 7.336% of the variance, and it was the least relevant factor of the five and was made up of two variables: age (0.834), and degree of satisfaction with the route taken (0.636).

As seen in Figure 8, the strongest relationship was in the category of interested in the ham (thicker blue circle), that they think ham has a good texture, that it is good for health, and that it has a good taste (thicker line of Figure 8, thicker grey and red circle, respectively), as well as lovers of ham, with respect to those initiated in ham. When not knowing the product, there were disparate opinions from which they think that the taste is very bad or that the taste is very good, as well as the texture or the opinion regarding whether it is healthy.

The results of the matrix showed that in the fourth component, “leisure and safety” presented a high load; the variables related to the route were valuations of the leisure offer, signalling, and safety of the route.

The fifth factor “satisfaction” consisted of two variables, namely, assessment of satisfaction with the route taken and age, with values higher than 0.8 for age and 0.6 for the degree of satisfaction.

Figure 9 shows the representation of each of the indicators of potentiality in the new factors created, so that each group of indicators of tourism potentiality was located around the axis that represents its typology of tourist resources. According to the results analysed, some items that responded to specific dimensions did not have such an important load in the factor analysis carried out, which does not mean that these were not influencing, so their relative importance in a future sample should continue to be evaluated.

Regarding the internal reliability index, Cronbach’s alpha was used, obtaining the value of 0.765 (Table 8), with the first dimension defining the supply, where there was a greater homogenization of 0.899, followed by the personal dimension with a 0.869.

The relationship was also analysed between the satisfaction of the ham tourist and age, and the statistic χ^2^ = 137,769, which was significant to 5%, which indicated that the variable satisfaction was related to age; as the person became older, the degree of satisfaction became greater.

There was also an association between the level of income and the degree of satisfaction with the ham route; χ^2^ = 292.99, which was significant at 5%. People with a higher level of income were unhappier with the Iberian ham route, mainly because they did not find adequate overnight accommodation, or the restaurant staff was not sufficiently qualified.

## 6. Discussion

The region of Andalusia, located in the south of Spain, has great gastronomic products, such as oil or ham, that are still not being sufficiently exploited from the tourism point of view despite having a great food quality endorsed by the PODs that exist in that region.

Ham is a healthy element of the Mediterranean diet, which is beginning to be known as a tourist product (ham tourism), although its life cycle is in its initial phase in Andalusia due to the small number of people who participate in it and the limited existing offerings compared to other types of gastronomic tourism, such as wine tourism [93], as there are no clearly defined ham routes.

Like other studies on gastronomic tourism [106,107], it is important to know the motivations of the ham tourist, as well as their profile, in order to know what the demand is. From the studied area, it was found that the profile of ham tourists who visit the PDOs in Andalusia is very similar to that of gastronomic tourists in Mealhada, Portugal [108], i.e., between 50 and 59 years, with medium education, and an upper middle income level between €1500 and €2000 per month, but in stark contrast to the demographics of the gastronomic tourists who visit Haiti, i.e.,. highly educated young people with a low income [105]. The profile of ham tourists in Andalusia is also different from that of the gastronomic tourists studied by Park [109], Robinson et al. [110], McKercher et al. [111], and Ignatov and Smith [112], who indicated that the tourists for whom gastronomy is a relevant component in the choice of a destination are approximately 45 years old and highly educated, married, and mostly Andalusian. Most of these tourists did not spend the night in the area, travelled with their family or friends, have a higher average income, and the average expenditure per day is higher than the sun and beach tourist [113]. Ham tourists have a high level of satisfaction of more than 75% with the sensory experience carried out because they are knowledgeable about the product they are going to visit and taste and value the experience very positively, since more than 98% would repeat the experience. These results are very similar to those obtained in other gastronomic products, such as oil [7] or wine [67,68], although some aspects must be improved, mainly those related to leisure and entertainment, since the offerings must be increased, and the information on the itineraries and the explanations of the guides of the routes given must be improved.

However, to make the ham route, there is a series of motivations in the tourist related to push factors, such as visiting pastures, visiting ham dryers, and enjoying the local gastronomy, or with the attraction (pull) factors, such as the leisure offering, security, the treatment received, etc. These result in the ham tourists being attracted to travel to Andalusia, the offering being the most important factor according to the factor analysis carried out, the results of which are in line with the studies of [114,115,116,117].

It is necessary to increase the tourism of ham in Andalusia so that businesses such as hotels and restaurants are profitable, especially in the areas of the region where the PODs are located, since most local tourists do not spend the night and hardly spend on provisions, except for in the purchase of Iberian products when they visit the dry seasons. It is necessary to promote two things: first, the gastronomy related to ham in exporting markets different from those of Spain, as is what happens in other Spanish communities such as Extremadura, where the gastronomic tourist spends the night [118]; and second, the pig and its habitat, since these are endogenous resources, rooted in tradition, with the purpose of attracting more visitors who know from pig breeding to food production and its enjoyment in order to promote local development as in other PODs, such as Guijuelo in Salamanca [119]. This would provide tourists not only with sensory experiences but with education in order to learn the traditions, such as the slaughter of the pig and manufacture of sausages, that in the last century were done at home, and to maintain customs that tourists can value in order to avoid their loss, establishing festivals where one can appreciate the uses and operations of culinary life of other times.

## 7. Conclusions

After the COVID-19 pandemic, it has been observed that tourists prefers less crowded places located mainly in areas in contact with nature and where they have more personalized attention, and that the product they receive is environmentally sustainable, the pastures being a visual paradise and the ham a product obtained from good breeding of the pig in that natural paradise, increasing the richness of Andalusian gastronomy.

Based on the results obtained where the five key factors of ham tourism have been determined, the most important factor being supply and within this accommodation and restaurant variables, we can therefore deduce that gastronomy is an essential motivation for travel. In addition, ham tourists think that this product is good for health, has a good texture, and tastes good, being a quality product that makes people have a very satisfactory sensory experience when they try it, valuing very positively the route of the ham or the POD. However, and according to the theory of the life cycle of the tourist product [120], it is in an initial phase of exploration, although with the advantage that more than 90% of tourists would repeat the experience.

Nevertheless, for ham to compete as an attractive tourist product in both national and international markets, a consistent and solid offering adapted to different motivations is needed, requiring private–public coordination in order to take measures and concrete actions to increase and promote gastronomic routes in Andalusia where they participate and make pairings between products such as wine and ham, or cheese and ham; however, to achieve this, it would be necessary to encourage the attraction of investors who are willing to create wealth and bet on gastronomy via tourism as a business opportunity.

In order to attract more ham tourists, the rural areas where Iberian ham is produced must adapt by diversifying tourism products (agrotourism, gastronomic tourism, sports tourism, etc.) and specialize their offerings to adjust to changes in consumer habits and meet their needs, which is ultimately the most important for loyalty and the attraction of new tourist consumers [41,121].

It can be concluded that the tourist who visits Andalusia for gastronomic reasons comprises only 10% of all tourists, since gastronomy is a complement to other motivations for travel and serves as positive lived experience. This tourism segment, considered strategic for both territorial and economic development, must be promoted and consolidated given its high added value. In short, oleo tourism, wine tourism, and ham tourism are tourist products with great growth potential that can favour the economic and territorial development of some Andalusian inland regions and constitute a quality tourist offering that responds to the expectations of visitors.

The results of this research can help both public administrations and private entities of Andalusia to become the first to create, improve, and promote rural development plans and promote aid for the development of tourism activity that is accessible to the small farmer who is dedicated to pig farming and to private entities for designing a tourism product based on the Iberian ham and tourist demand. The objective of these efforts would be to improve the tourist offerings for this very unique element and to transform ham tourism into a type of special interest tourism as classified by [122]. To increase the amount of Iberian ham in Andalusia, a good marketing campaign in international markets would be needed first to publicize the gastronomic product, which is unknown by many tourists because it does not exist in their country. In addition, foreign tourists have greater purchasing power than do national tourists and are able to spend more in the area, generating more income for producers. As indicated by Pulido et al. (2021) [123], ham tourists with professional purposes show a special interest in companies such as international distributors or restaurants that can supply their products at the international scale and/or continuously across time.

For future lines of research, this study could be carried out in other destinations in Spain, such as Estremadura, or Castille and Leon, and the results obtained in this work could be compared with those of other destinations. Another possible line of research could be to perform this same study in Andalusia, but aimed at international tourists, in order to examine their motivations and thus establish a segmentation of the touristic offerings of the community according to the type of tourist, national or international. Another possible line of research would be to compare the profile of the ham tourist with that of other products, such as wine or oil, in order to create combined gastronomic routes of various products based on similar tourist profiles.

## Figures and Tables

**Figure 1 foods-11-02277-f001:**
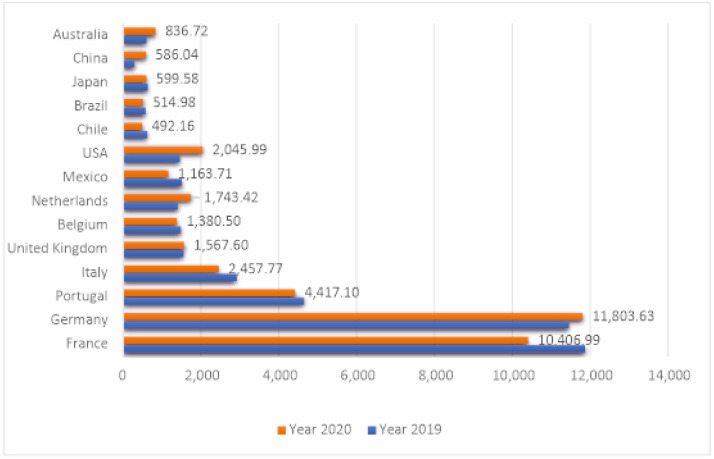
Export of ham (tons). Source: own elaboration based on information from the Ministry of Agriculture, Fisheries, and Food (MAPA) [9].

**Figure 2 foods-11-02277-f002:**
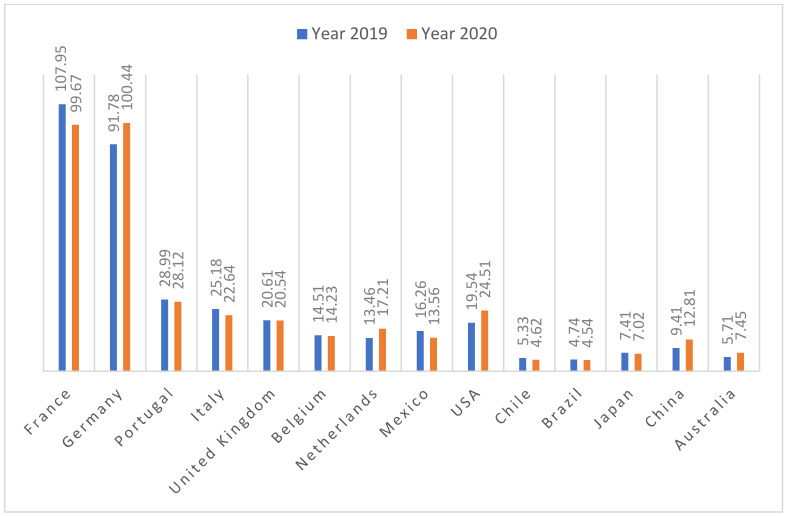
Export of ham (million euros). Source: own elaboration based on information from the Ministry of Agriculture, Fisheries, and Food (MAPA) [9].

**Figure 3 foods-11-02277-f003:**
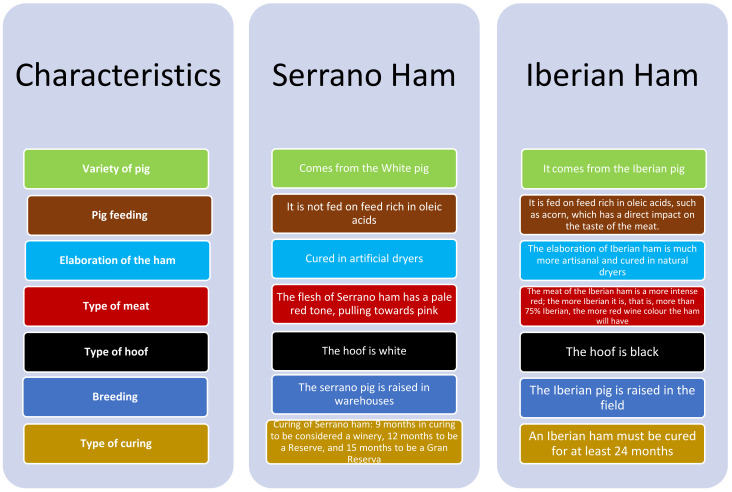
Differences between Serrano ham and Iberian ham. Source: own elaboration.

**Figure 4 foods-11-02277-f004:**
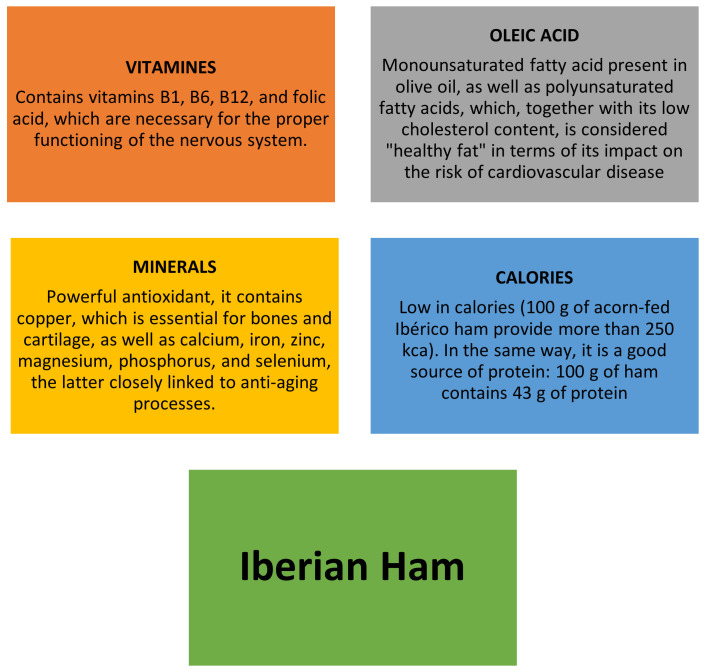
Benefits of Serrano ham. Source: own elaboration from Amaya-Corchuelo et al. [11] and Pérez et al. [12].

**Figure 5 foods-11-02277-f005:**
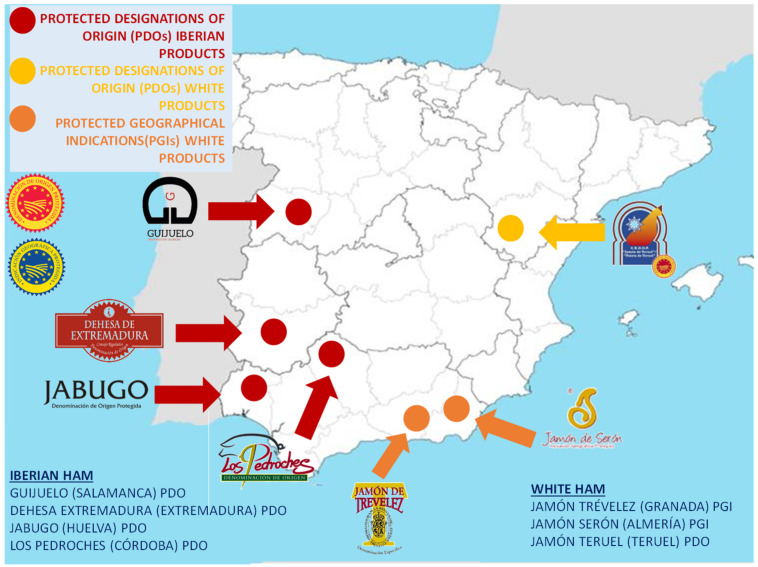
PDO and PGI of ham of Spain. Source: own elaboration.

**Figure 6 foods-11-02277-f006:**
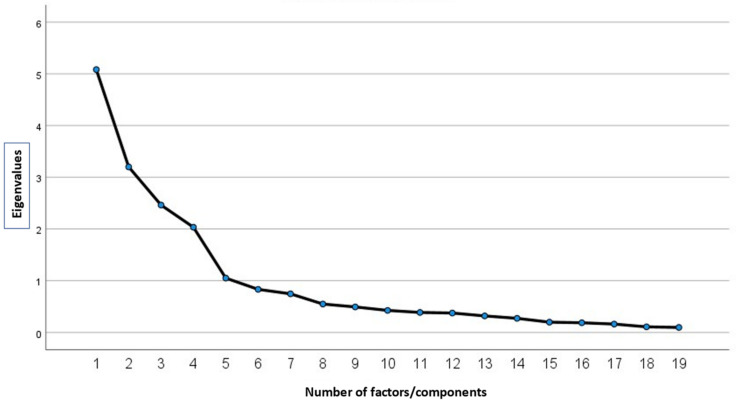
Result of the sedimentation of factors.

**Figure 7 foods-11-02277-f007:**
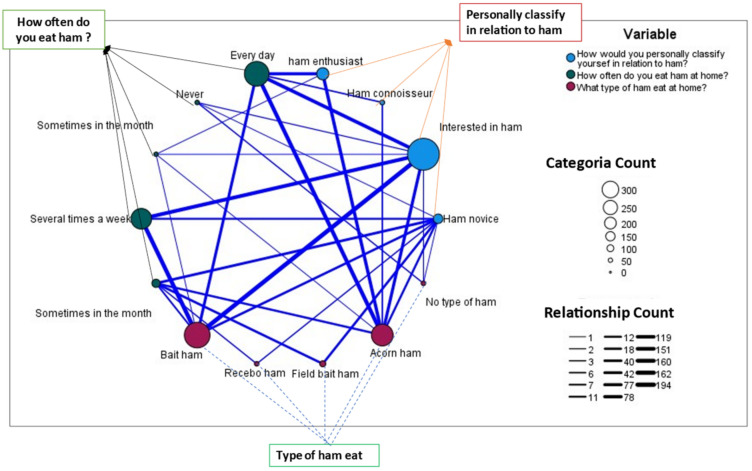
Map of ham tourist–ham–ham-type consumption relationships.

**Figure 8 foods-11-02277-f008:**
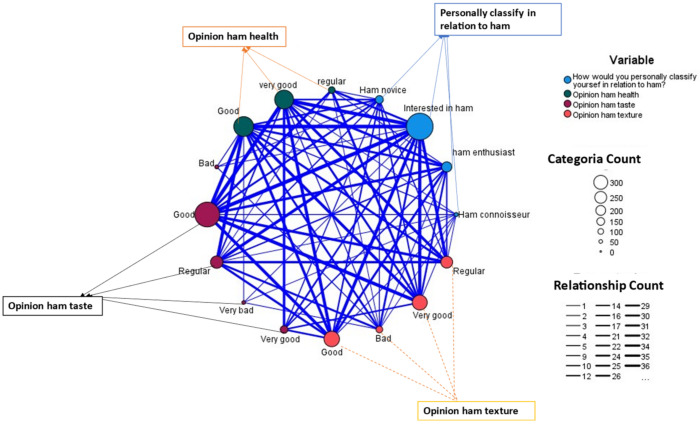
Map of relationships valuation of ham–texture–taste–health with respect to the classification of the ham tourist.

**Figure 9 foods-11-02277-f009:**
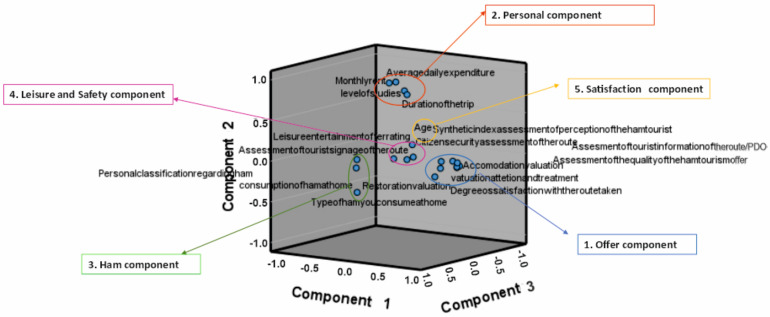
Component chart in rotated space.

**Table 1 foods-11-02277-t001:** PDO and PGI from Spain.

Product	PDO		PGI	
	Spain	Andalusia	Spain	Andalusia
Food	102	20	90	13
Wine	97	8	42	16
Spirits			19	1
Aromatised wine products			1	1
TOTAL	199	28	152	31

Source: own elaboration based on information from the Ministry of Agriculture, Fisheries, and Food (MAPA) [9].

**Table 2 foods-11-02277-t002:** Survey data sheet.

	Demand Survey
Population	Tourists of both sexes over 18 years old who made/visited a route of Iberian ham/POD of Andalusia
Sample size	409
Sampling error	±4.1%
Trust level	95%; p = q = 0.5
Sampling system	Simple random
Date of fieldwork	September 2021–January 2022

**Table 3 foods-11-02277-t003:** Profile of the ham tourist in Andalusia.

Block	Question	Classification	Percentage
**Personal characteristics of the ham tourist**	**Age**	18–29 years old	14.2
		30–39 years old	27.1
		40–49 years old	20.3
		**50–59 years old**	**31.1**
		Over 60 years old	7.3
	**Educational level**	No completed studies	9.3
		Primary and secondary education	19.6
		**Secondary/VET studies**	**43.8**
		Higher education	27.4
	**Gender**	**Man**	**57.5**
		Woman	42.5
	**Marital status**	Single	26.9
		**Married**	**47.7**
		Divorced/separated	25.2
		Other	0.2
	**Income level of the family unit**	Less than 1000 euros	19.8
		1001–1500 euros	19.8
		**1501–2000 euros**	**30.3**
		2001–2500 euros	20.0
		+ 2500 euros	10
	**Who are you doing the tour with?**	Alone	3.2
		**Accompanied by my partner**	**48.9**
		With friends	37.7
		With relatives	10.3
	**Where are you from?**	**Andalusia**	**58.9**
		Rest of Spain (except Andalusia)	30.1
		European Union (except Spain)	10.0
		United States	0.2
		Rest of the world (except United States)	0.7
	**Employment situation**	Self-employed	10.2
		**Salaried employee**	**61.4**
		Retired	17.4
		Unemployed	8.3
		Student	2.7
	**Duration of trip**	**Less than 24 h**	**53.5**
		1–3 days	34.5
		More than 3 days	12.0
	**Daily spending**	Less than 30 euros	11.2
		30–65 euros	24.0
		**66–100 euros**	**42.1**
		More than 100 euros	22.7
**Questions about the visit**	**Number of people who have accompanied you on the tour**	1 person	13.4
		**2 to 4 people**	**65.3**
		More than 4 people	21.3
	**Has the PDO or the culinary tour met your expectations?**	**Yes**	**79.2**
		No	20.8
	**What do you think could be done to improve the tour?**	No answer	0.0
		**Signage**	**49.1**
		Explanation of the tour or PDO	40.3
		More audio-visual media	10.3
		Other	0.2
	**Would you be interested in receiving more information after the visit?**	**Yes, if it is free**	**59.7**
		Yes, in any case	30.3
		I do not consider it necessary	10.0
	**Did you come here specifically to do the ham tour or was it offered to you when you were already in Andalusia?**	**I specifically came here to do the tour**	**59.7**
		It was offered to me by chance	40.3
	**Does the price paid for the tour seem good value for money?**	**Yes**	**96.8**
		No	3.2
	**How did you find out about the tour?**	Travel agency	13.2
		**Online, through social networks**	**47.9**
		On the recommendation of friends and family	32.8
		Other media	6.1
	**I would repeat this experience with a similar tour**	**Yes**	**98.3**
		No	1.7
	**Degree of satisfaction with the visit**	Less than 25%	0.7
		25–50%	1.5
		51–75%	3.7
		**76–99%**	**55.7**
		100%	38.4
**Questions about the motivation for the visit**	**What is your main motivation for the visit?**	Finding out about the culinary tradition of the place	39.6
		**Learning about the ham process, visiting meadows (dehesa)**	**50.4**
		Attending ham festivals	10.0
	**How do you assess the current situation in terms of tourism management of places like the ones you have visited?**	**Good**	**52.3**
		Regular	27.1
		Bad	20.5
	**What do you think about the creation of a combined tour of various gastronomic products with theatrical representation?**	**I agree**	**97.3**
		I do not agree; I prefer to do a single culinary tour rather than several	2.7
**Questions about ham consumption**	**How often do you eat ham at home?**	**Every day**	**49.4**
		Several times a week	39.6
		Once a fortnight	0.5
		Sometimes in the month	9.8
		Never	0.7
	**What type of ham do you eat at home?**	Iberian ham	42.3
		Recebo ham	**51.8**
		Cebo ham	4.4
		Field cebo ham	0.7
		No type of ham	0.7
	**How would you personally classify yourself in relation to ham?**	Ham connoisseur	1.5
		Ham enthusiast	19.1
		**Interested in ham**	**66.7**
		Ham novice	12.7
Opinions regarding ham	Is ham healthy?	Very good	44.0
		**Good**	**46.9**
		Regular	9.0
		Bad	0.0
		Very bad	0.0
	The texture of the ham?	Very good	32.8
		**Good**	**34.5**
		Regular	23.7
		Bad	9.0
		Very bad	0.0
	The taste of ham?	Very good	11.7
		**Good**	**61.4**
		Regular	25.4
		Bad	1.0
		Very bad	0.5

**Table 4 foods-11-02277-t004:** KMO and Bartlett test.

Kaiser–Meyer–Olkin sampling adequacy measure		0.75385
Bartlett sphericity test	Approx. Chi-square	4869.413
	Gl	171
	Itself	0.000

**Table 5 foods-11-02277-t005:** Communalities. Extraction method: principal component analysis.

	Initial	Extraction
Age	1.000	0.784
Level of studies	1.000	0.716
Monthly rent	1.000	0.888
Duration of the trip	1.000	0.617
Average daily expenditure	1.000	0.846
Degree of satisfaction with the route taken	1.000	0.734
Consumption of ham at home	1.000	0.833
Type of ham consumed at home	1.000	0.773
How it is classified with respect to the knowledge/taste/use of ham	1.000	0.781
Accommodation review	1.000	0.797
Restoration valuation	1.000	0.780
Leisure/entertainment offer rating	1.000	0.692
Assessment of the quality of the ham tourism offer	1.000	0.792
Evaluation attention and treatment received during the route	1.000	0.626
Assessment of tourist signage of the route/PDO	1.000	0.589
Assessment of tourist information of the route/PDO	1.000	0.587
Assessment of cultural heritage municipality of the route	1.000	0.85
Citizen security assessment of the route	1.000	0.595
Synthetic index assessment of perception of the ham tourist	1.000	0.614

**Table 6 foods-11-02277-t006:** Total variance explained.

Component	Initial Eigenvalues			Sums of Loads Squared from the Extraction			Sums of Charges Squared in the Rotation		
	Total	% Variance	Cumulative %	Total	% Variance	Cumulative %	Total	% Variance	Cumulative %
1	5.083	26.752	26.752	5.083	26.752	26.752	4.671	24.583	24.583
2	3.199	16.839	43.591	3.199	16.839	43.591	3.084	16.233	40.816
3	2.461	12.954	56.545	2.461	12.954	56.545	2.381	12.529	53.345
4	2.033	10.702	67.247	2.033	10.702	67.247	2.299	12.099	65.444
5	1.051	5.533	72.781	1.051	5.533	72.781	1.394	7.336	72.781

Extraction method: principal component analysis.

**Table 7 foods-11-02277-t007:** Component matrix rotated to.

	Component				
	1	2	3	4	5
Accommodation Review	0.889	0.027	0.025	−0.026	0.061
Assessment of the quality of the ham tourism offer	0.889	−0.020	−0.020	0.23	−0.010
Restoration valuation	0.875	−0.027	0.014	0.063	0.103
Assessment of cultural heritage municipality of the route	0.835	−0.048	−0.044	−0.255	0.138
Assessment of tourist information of the route/PDO	0.704	0.004	−0.143	0.241	−0.111
Evaluation attention and treatment received during the route	0.695	−0.050	0.104	0.277	0.231
Synthetic index assessment of perception of the ham tourist	0.629	0.012	−0.006	0.462	−0.070
Monthly Rent	−0.041	0.931	0.135	0.021	0.031
Average daily expenditure	−0.059	0.917	−0.030	−0.020	0.006
Level of studies	−0.016	0.795	−0.151	−0.005	−0.248
Duration of the trip	0.033	0.755	−0.135	−0.032	0.164
Consumption of ham at home	0.011	0.002	0.906	−0.032	−0.107
How it is classified with respect to the knowledge/taste/use of ham	−0.028	0.091	0.835	−0.017	−0.272
Type of ham you consume at home	−0.037	−0.316	0.818	−0.045	0.016
Leisure/entertainment offer rating	0.124	−0.015	−0.004	0.819	0.070
Assessment of tourist signage of the route/PDO	−0.073	−0.025	−0.005	0.758	0.090
Citizen security assessment of the route	0.168	0.011	−0.073	0.748	−0.042
Age	0.015	0.118	−0.272	−0.014	0.834
Degree of satisfaction with the route taken	0.443	−0.214	−0.138	0.262	0.636

Extraction method: principal component analysis. Rotation method: Varimax with Kaiser normalization. The rotation converged in 5 iterations.

**Table 8 foods-11-02277-t008:** Reliability statistics.

	Cronbach’s Alpha	No. of Elements
Total	0.765	19
Offer	0.899	7
Personal	0.869	4
Ham	0.826	3
Leisure and Security	0.660	3
Satisfaction	0.506	2

## Data Availability

The data presented in this study are available on request from the corresponding author.

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
