# Peer review of "Ham Tourism in Andalusia: An Untapped Opportunity in the Rural Environment"

_foods, 2022, doi:10.3390/foods11152277_

Round 1
Reviewer 1 Report
The subject of the article is interesting and it is, somehow, linked to the objectives of the journal). However, there are a number of issues that have to be reconsidered.
For better visibility on databases, the authors are asked not to repeat among keywords the words/concepts included in the title of the article.
The introduction is too long. This should not be a review of research, but an introduction to the topic of your study Propose only to focus on the main subject. Alos, the presentation of the structure of the paper is missing, as the main and secondary objectives.
In the methodology, there is necessary to explain in more detail how the sample was selected and why the sample is representative of the entire studied population.
lines 333-385. These are not results and/or discussions of results, there are more Introductive parts.
Lines 386-436. These are not results and/or discussions of results, there are more methodological parts.
line 484. Extraction of components. Is that a sub-chapter?
Conclusions and Discussions could be split into 2 distinct parts; the Conclusions. Limits, possibilities of study follow-up, and recommendations for scholars, companies, tourism facilitators, and government must be clearer.
Author Response
Dear Reviewer:
We appreciate all your comments and suggestions you have made about the content of the paper. We have tried to address all of them.
The subject of the article is interesting and it is, somehow, linked to the objectives of the journal). However, there are a number of issues that have to be reconsidered.
For better visibility on databases, the authors are asked not to repeat among keywords the words/concepts included in the title of the article.
You are right. Entering different words in the title and in the keywords can improve the search for the paper in metasearch engines and internet databases. In this sense, we have changed the keyword from the concept of ham tourism to that of gastronomic tourism.
The introduction is too long. This should not be a review of research, but an introduction to the topic of your study Propose only to focus on the main subject. the presentation of the structure of the paper is missing, as the main and secondary objectives.
Again, you are right. We have reduced the length of the introduction and added the structure of the work and the objectives.
In the methodology, there is necessary to explain in more detail how the sample was selected and why the sample is representative of the entire studied population.
The sample was selected by a simple random sampling method indicated in table 2 to people who visited an Iberian ham route or a PDO (Protected Designation of Origin) of Andalusian Iberian Ham. The formula used for the sample size is that of infinite population or unknown
lines 333-385. These are not results and/or discussions of results, there are more Introductive parts.
Lines 386-436. These are not results and/or discussions of results, there are more methodological parts.
line 484. Extraction of components. Is that a sub-chapter?
It is true that the text, reading your comments, could be organized in a much clearer way. Therefore, following your suggestions we have moved the content of lines 333-385 to a new section of Literature Review. On the other hand, the content of lines 386-436 has been moved to the Methodology section, which is where it fits better. Finally, we have removed line 484 as it suggested a new heading when it was not.
Conclusions and Discussions could be split into 2 distinct parts; the Conclusions. Limits, possibilities of study follow-up, and recommendations for scholars, companies, tourism facilitators, and government must be clearer.
Two sections have been created, one for discussion and the other for conclusions. In the latter, some strategies for companies and academics have been indicated and the limitations of the study have been added.
Reviewer 2 Report
Thank you very much for giving me the opportunity to review this interesting manuscript.
I have enjoyed reading about Iberico ham and learning so much about its importance and differences with serrano ham.
Now, I think the study is interesting and has potential for readers, however I find several shortcomings or weaknesses. The study has to move from a case study to a scientific study.
In other words, your presentation can be improved by restructuring a large part of the manuscript.
1.- Introduction: There is no theory, there is better an excellent explanation of the ham. The objectives of the study are not clear. What contribution to academic literature will they make? Please do not focus on the ham, expose important parts of the theory to be treated and justify the study based on this theory.
2.- Literature: There is no literature on the study only on ham. Please do not focus only on the ham. For example, what previous studies of gastronomic routes exist? What studies on visits, ham tour packages exist? On the behavior of the tourist in a gastronomic route. There is no contribution to the academic literature because I have not found a literature on which they are based. Therefore, there are no theoretical implications in the present study.
3.- Methodology: It took a long time to collect this sample, please explain why. What problems did they have? The questionnaire from which studies does it come from? Detail its parts and how each question or section was measured.
4. Results: Here is literature. Literature exists where it shouldn't exist. Based on your results, which are good, propose previous literature in the literature section. In other words, put previous findings of what you found in this study.
5.- Discussion: Does not exist: this section must be created: expose your findings with the previous findings. Expose your contribution to the academic literature. Discuss practical implications of your findings.
6.- Conclusions: Propose your contribution to the literature through theoretical implications. Limitations and future lines of research.
I ask you to restructure the manuscript. I wish you the best of luck.
Author Response
Dear Reviewer:
We appreciate all your comments and suggestions you have made about the content of the paper. We have tried to address all of them.
Thank you very much for giving me the opportunity to review this interesting manuscript.
I have enjoyed reading about Iberico ham and learning so much about its importance and differences with serrano ham.
Now, I think the study is interesting and has potential for readers, however I find several shortcomings or weaknesses. The study has to move from a case study to a scientific study.
In other words, your presentation can be improved by restructuring a large part of the manuscript.
1.- Introduction: There is no theory, there is better an excellent explanation of the ham. The objectives of the study are not clear. What contribution to academic literature will they make? Please do not focus on the ham, expose important parts of the theory to be treated and justify the study based on this theory.
The introduction has been modified by adding the structure and objectives of the work, as well as the contribution of this work to the scientific literature.
2.- Literature: There is no literature on the study only on ham. Please do not focus only on the ham. For example, what previous studies of gastronomic routes exist? What studies on visits, ham tour packages exist? On the behavior of the tourist in a gastronomic route. There is no contribution to the academic literature because I have not found a literature on which they are based. Therefore, there are no theoretical implications in the present study.
Following your suggestions, we have created a new literature review section where the research on gastronomy and gastronomic routes that serve as the basis for the development of our study have been commented.
3.- Methodology: It took a long time to collect this sample, please explain why. What problems did they have? The questionnaire from which studies does it come from? Detail its parts and how each question or section was measured.
The sample was taken from September 2021 to January 2022, as indicated in Table 1, with two types of questions in the study: nominal qualitative questions such as gender, ordinal qualitative questions such as educational level, etc., or untabulated quantitative questions such as educational level. average daily income expenditure, or Likert scale (1-10) as an assessment of the route (restoration, security and treatment received, heritage, etc...)
- Results: Here is literature. Literature exists where it shouldn't exist. Based on your results, which are good, propose previous literature in the literature section. In other words, put previous findings of what you found in this study.
Some lines have been changed. We have moved the content of lines 333-385 to a new section of Literature Review. On the other hand, the content of lines 386-436 has been moved to the Methodology section, which is where it fits better
5.- Discussion: Does not exist: this section must be created: expose your findings with the previous findings. Expose your contribution to the academic literature. Discuss practical implications of your findings.
A Discussion Section has been created where the results obtained are checked and compared with other findings from similar studies carried out by other authors.
6.- Conclusions: Propose your contribution to the literature through theoretical implications. Limitations and future lines of research.
We have worked on a new Conclusions section, and we believe that this section has now been improved, pointing out the limitations of the study and proposing new lines of research.
Reviewer 3 Report
General comments
The study aimed to know the factors that influence the development of ham tourism in Andalusia.
The major strengths of this study are that a higher level of knowledge about ham tourists is a relevant strand of research for creating and implementing ad hoc strategies among Andalusia's firms. From this perspective, the paper assessed key points that deserve attention in this particular tourism segment.
The paper is quite well developed, with a proper analysis method corroborated by an interesting discussion. However, I guess there is still room for improvement, especially in the presentation of results and concluding sections, which have been thoroughly listed below.
Originality
The paper demonstrates significant elements of originality.
Presentation of productions from the study area
· On page 3 (lines 117 and 119), there is a repetition concerning the 'Ministry of Agriculture, Fisheries and Food (MAPA)'.
· The description of market shares and the comments on long-term changes (2019 over 2010) are of interest for interpreting the evolution of the sector (page 4, lines 132-136).
· At the end of page 5, there seems to be a statement that remains isolated.
Tables and Figures:
· One can believe that Figure 1 should be presented more efficiently and in tabular form by adding an initial column covering the characteristics considered and then in columns 2 and 3 reporting only the differences.
· In Figure 3, round off the data shown in the graph and use a thousand separators. The heading United Country (...) is incomplete.
· In Figure 4, round the decimal places in a homogenous manner. In addition, please use lower and upper case letters.
· In Table 3, 'one' is shown in gender instead of 'man'. It should be better to use impersonal forms' questions'.
· The definition of levels in 'Degree of satisfaction with the visit' does not appear clear.
· Table 6 has axis titles in Spanish.
· Table 8 has coloured highlights.
· I think a more effective visual representation in terms of interpretation should be chosen from figures 7 to 9. It is believed there are too many charts and tables and a bit rushed in work presented.
Methodology
The methodology used is appropriate.
Results
The analysis of the results should start on page 12 (line 451 onwards); the rest seems more logically ascribable to the methodology and should be summarised in parts (e.g. lines 386-436).
Conclusions
· Page 11, line 377. It is assumed that four decimal places are not needed here (3.1958).
· The conclusions should move to an interpretative level that recalls the consistency between the objectives, method and results of the paper without recalling the statistical tests again.
Relationship to Literature
The literature review is reasonable. However, one can believe that considerations relating to the development phase of (ham and other gastronomic products) tourism is a key concept of particular interest, a context in which the characteristics of business supply can play a strategic role. In this perspective, some studies have placed particular emphasis on these aspects and, therefore, should be included in the bibliography:
· Boatto, V., Galletto, L., Barisan, L., & Bianchin, F. (2013). The development of wine tourism in the Conegliano Valdobbiadene area. Wine Economics and Policy, 2(2), 93-101.
· Romão, J., Guerreiro, J., & Rodrigues, P. (2013). Regional tourism development: Culture, nature, life cycle and attractiveness. Current Issues in Tourism, 16(6), 517-534.
· Moscardo, G., Minihan, C., & O’Leary, J. (2015). Dimensions of the food tourism experience: Building future scenarios. The future of food tourism: Foodies, experiences, exclusivity, visions and political capital, 71, 208.
Limitations and Future Research
The study's limitations should be mentioned more clearly in the concluding section.
Quality of Communication
I guess that the effort produced by the authors in this interesting field of research can be recognized after a major review and recommended improvements mainly to the results and concluding sections.
Final comments to the Authors
It is believed that the effort put in by the authors on this interesting topic of ham tourism segment could be recognized for publication in Food but should be improved to make it more effective after a major revision.
Author Response
The study aimed to know the factors that influence the development of ham tourism in Andalusia.
The major strengths of this study are that a higher level of knowledge about ham tourists is a relevant strand of research for creating and implementing ad hoc strategies among Andalusia's firms. From this perspective, the paper assessed key points that deserve attention in this particular tourism segment.
The paper is quite well developed, with a proper analysis method corroborated by an interesting discussion. However, I guess there is still room for improvement, especially in the presentation of results and concluding sections, which have been thoroughly listed below.
Originality
The paper demonstrates significant elements of originality.
Presentation of productions from the study area
- On page 3 (lines 117 and 119), there is a repetition concerning the 'Ministry of Agriculture, Fisheries and Food (MAPA)'.
The source 'Ministry of Agriculture, Fisheries and Food (MAPA)' has been repeated because line 117 MAPA refers to PDOs and line 119 refers to sales of agri-food products. That is, it is the same source for two different concepts, so we understood that it was better to put it twice.
- The description of market shares and the comments on long-term changes (2019 over 2010) are of interest for interpreting the evolution of the sector (page 4, lines 132-136).
It has been indicated in this way because the authors think that there has been this evolution.
- At the end of page 5, there seems to be a statement that remains isolated.
This statement has been removed because it corresponds to the title of the figure
Tables and Figures:
- One can believe that Figure 1 should be presented more efficiently and in tabular form by adding an initial column covering the characteristics considered and then in columns 2 and 3 reporting only the differences.
Following your suggestions, we have changed Figure 1 by adding a new column with the characteristics considered in the study.
- In Figure 3, round off the data shown in the graph and use a thousand separators. The heading United Country (...) is incomplete.
Indeed, there was a material error in this figure. It has been corrected by putting the full name of the United Kingdom and changing the commas to periods.
- In Figure 4, round the decimal places in a homogenous manner. In addition, please use lower and upper case letters.
It has been homogenized by putting two decimals and only the initial of the country has been capitalized to be homogeneous with figure 2.
- In Table 3, 'one' is shown in gender instead of 'man'. It should be better to use impersonal forms' questions'.
It was a transcription error. It is already corrected.
- The definition of levels in 'Degree of satisfaction with the visit' does not appear clear.
- Table 6 has axis titles in Spanish.
The degree of satisfaction is related to their expectations about the visit. For instance, 0.7 of the respondents only felt their expectations were met by 25%, while 38.4% considered their expectations were met 100% and are very satisfied. This has been clarified in the text. The value 94.1 that appears on line 470 is the sum of the degrees of satisfaction greater than 75% (55.7+38.4).
- Table 8 has coloured highlights.
The variables that are part of each component have been indicated in color.
- I think a more effective visual representation in terms of interpretation should be chosen from figures 7 to 9. It is believed there are too many charts and tables and a bit rushed in work presented.
We have tried to better explain the content of the figures in the text. However, we cannot modify the figures themselves because they are the ones that allow the SPSS program used in this work to be carried out.
Methodology
The methodology used is appropriate.
Results
The analysis of the results should start on page 12 (line 451 onwards); the rest seems more logically ascribable to the methodology and should be summarised in parts (e.g. lines 386-436).
Indeed, you are right. Following your suggestions and those of another reviewer, we have proceeded to reorganize all the work. The first lines of results have been changed to a new Literature Review section (lines 33-385) and lines 346-451 have been placed in the Methodology section. Likewise, we have passed the content of 452 onwards to Results.
Conclusions
- Page 11, line 377. It is assumed that four decimal places are not needed here (3.1958).
- The conclusions should move to an interpretative level that recalls the consistency between the objectives, method, and results of the paper without recalling the statistical tests again.
The conclusions have been modified according to the reviewer's suggestions in a more interpretive way
Relationship to Literature
The literature review is reasonable. However, one can believe that considerations relating to the development phase of (ham and other gastronomic products) tourism is a key concept of particular interest, a context in which the characteristics of business supply can play a strategic role. In this perspective, some studies have placed particular emphasis on these aspects and, therefore, should be included in the bibliography:
- Boatto, V., Galletto, L., Barisan, L., & Bianchin, F. (2013). The development of wine tourism in the Conegliano Valdobbiadene area. Wine Economics and Policy, 2(2), 93-101.
- Romão, J., Guerreiro, J., & Rodrigues, P. (2013). Regional tourism development: Culture, nature, life cycle and attractiveness. Current Issues in Tourism, 16(6), 517-534.
- Moscardo, G., Minihan, C., & O’Leary, J. (2015). Dimensions of the food tourism experience: Building future scenarios. The future of food tourism: Foodies, experiences, exclusivity, visions and political capital, 71, 208.
A specific Literature Review section has been created, adding studies on gastronomy, gastronomic routes, and in which the authors suggested by the reviewer have been considered.
Limitations and Future Research
The study's limitations should be mentioned more clearly in the concluding section.
In the Conclusions section, a paragraph with the limitations of the study has been added.
Quality of Communication
I guess that the effort produced by the authors in this interesting field of research can be recognized after a major review and recommended improvements mainly to the results and concluding sections.
Final comments to the Authors
It is believed that the effort put in by the authors on this interesting topic of ham tourism segment could be recognized for publication in Food but should be improved to make it more effective after a major revision.
Round 2
Reviewer 1 Report
The authors succeeded in answering my concerns.
Author Response
Dear reviewer,
Thank you for recognizing our effort. We have tried to respond to all the requirements of the reviewers, so we have improved the English version of our work, the literature review, and the methodology section.
Reviewer 2 Report
The manuscript has improved remarkably.
I believe that it will improve much more if a reflection is added at the end of the Literature on the results of the previous findings in relation to the analyzed topic.
I do not find in the manuscript from which previous studies the questionnaire originates. As they demonstrate to the readers that the questionnaire is scientific and validated and acts to provide reliable findings.
Author Response
Dear reviewer,
Thank you for recognizing our effort. We have tried to respond to all the requirements of the reviewers, so we have improved the English version of our work, the literature review, and the methodology section.
I do not find in the manuscript from which previous studies the questionnaire originates. As they demonstrate to the readers that the questionnaire is scientific and validated and acts to provide reliable findings.
Following your suggestions, we have added some papers where the validity of our questionnaire has been demonstrated.
Reviewer 3 Report
On the whole, the corrections made by the authors seem satisfactory, albeit with a few oversights that could be incorporated into the final version of the paper (e.g. references in the literature supporting the importance of the analysis of the development phase of gastronomic tourism).
Author Response
Dear reviewer,
Thank you for recognizing our effort. We have tried to respond to all the requirements of the reviewers, so we have improved the English version of our work, the literature review, and the methodology section.
On the whole, the corrections made by the authors seem satisfactory, albeit with a few oversights that could be incorporated into the final version of the paper (e.g. references in the literature supporting the importance of the analysis of the development phase of gastronomic tourism).
Some previous works on the life cycle phase of gastronomy tourism have been added and the English version of the paper has been revised.